# Edible Coating Based on Carnauba Wax Nanoemulsion and *Cymbopogon martinii* Essential Oil on Papaya Postharvest Preservation

Josemar Gonçalves de Oliveira Filho [1], Guilherme da Cruz Silva [2], Fernanda Campos Alencar Oldoni [1], Marcela Miranda [1], Camila Florencio [3], Raissa Moret Duarte de Oliveira [2], Mariana de Paula Gomes [2] and Marcos David Ferreira [3,*]

[1] School of Pharmaceutical Sciences, São Paulo State University (UNESP), Araraquara 14800-903, Brazil
[2] Department of Biotechnology, Federal University of São Carlos, São Carlos 13565-905, Brazil
[3] Brazilian Agricultural Research Corporation, Embrapa Instrumentation, São Carlos 70770-901, Brazil
* Correspondence: marcos.david@embrapa.br

**Abstract:** Papaya is a fruit of great importance worldwide. However, significant losses during postharvest have been reported, which can be minimized by applying lipid nanoemulsions as edible coatings associated with natural antimicrobial compounds. These coatings provide a barrier to gases and water vapor, in addition to improving mechanical properties, thereby delaying natural senescence and minimizing deterioration by phytopathogens during storage. The aim of this study was to investigate the preservation potential of papaya fruits treated with an edible coating based on an association between carnauba wax nanoemulsion (CWN) and *Cymbopogon martinii* essential oil (CEO). Coatings formulated with CWN and/or CEO were applied to papaya fruits, and resulted in late ripening during the 12-day storage period, without negatively affecting postharvest fruit quality parameters. The coatings reduced weight loss and maintained firmness, in addition to delaying changes in fruit color during storage. Coatings formulated with CWN + CEO were the most efficient formulations in reducing the incidence and severity of fruit rots during storage. CWN coatings incorporating CEO present additional functionalities in maintaining postharvest quality parameters of papaya fruits.

**Keywords:** papaya; shelf life; nanotechnology; postharvest diseases

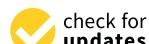



## 1. Introduction

Papaya (*Carica papaya* L.) is an important fruit for agriculture in tropical and subtropical regions. For being a climacteric fruit, it matures relatively quickly and is susceptible to many adversities in postharvest, which makes it difficult to maintain quality during storage [1,2]. Factors such as inadequate temperature and incidence of microbial diseases make the shelf life of papaya very brief, causing losses to producers, distributors, and consumers [2,3]. Some traditional technologies aim to overcome this problem, such as ozone radiation, modified atmosphere packaging, preservatives, and irradiation [4]. However, costs and limitations imposed by these methods imply the search for new alternatives.

In recent decades, consumers' demand for safer and more sustainable options has significantly impacted the food industry [5]. In this context, the postharvest of fruits and vegetables stands out for the use of materials based on non-biodegradable plastics, which causes their undue disposal and accumulation in nature [6]. Therefore, there is currently an increased interest in research and development of technologies that are capable of reducing food losses during storage, while making use of biodegradable, healthy, and safe products [7].

Edible coatings are primarily made up of materials of natural origin that protect, maintaining the quality and extending the shelf life of fruits [8,9]. The classification of edible

coatings varies according to their main component, which are usually polysaccharides, proteins, or lipids, alone or in combination [10]. These coatings are applied to the surface of the fruit through dipping, brushing, or spraying [11,12], and act as a barrier that limits the passage of volatile compounds and regulates the effects of atmospheric changes, in addition to being able to reduce harmful chemical reactions and provide protection against diseases caused by microorganisms [13–15].

Carnauba wax nanoemulsion (CWN) is an edible lipid coating that has been shown to be promising in postharvest application of fruits [16–18]. Carnauba is extracted from palm tree leaves from *Copernicia prunifera*, native to northeastern Brazil [19]. It has been demonstrated its potential for papaya [17,18] in terms of delaying ripening, reducing both the loss of fresh mass and appearance of diseases during storage. Furthermore, carnauba wax helps improve the visual appearance of the product, giving it more shine [17,20]. In a nanoemulsion, the sizes of the suspended particles are within the nanometric scale and it provides a series of advantages, such as greater dispersion across cell membranes, maximization of physicochemical and biological properties, better stability, and homogeneity [20].

*Cymbopogon martinii* essential oil (CEO), also known as palmarosa tree, is a great candidate for use in the control of microbial diseases. Some studies have already proven its bioactivity against pathogenic fungi and bacteria [21–25], which is mainly due to the high concentration of geraniol present in its composition, a compound whose lipophilic characteristic is directly related to adhesion and destruction of microbial membranes [22,25]. Therefore, the aim of this study was to investigate the preservation potential of papaya fruits treated with an edible coating based on an association between carnauba wax nanoemulsion and *C. martinii* essential oil.

## 2. Materials and Methods

### 2.1. Materials

Essential oil of palmarosa (*Cymbopogon martinii*) lot 2489 was purchased from Laszlo Aromaterapia (Belo Horizonte, Minas Gerais, Brazil). The chemical profiles of CEO were determined using gas chromatography and mass spectrometry (GC-MS, QP-5000, Shimatzu, Columbia, MD, USA) previously published by our research group (de Oliveira Filho et al., 2020). The major volatile contents (i.e., ≥1%) of *C. martinii* essential oil were geraniol (83.82%), geranyl acetate (7.49%), linalool (2.48%), and caryophyllene (1.33%). Papaya fruits, solo group, cultivar Golden, were carefully shipped from a commercial farm (Mucuri, Brazil) to the postharvest laboratory, Embrapa Instrumentação, São Carlos, São Paulo State-Brazil, and sanitized with specific detergent for fruits and chlorine dioxide (200 ppm). They were then selected by lacking standard defects, size, and maturity stage (stage 1 of maturation, less than 15% of skin surface covered by a yellow color) [26].

### 2.2. Nanoemulsions Production

A carnauba wax nano emulsion (CWN) was formulated according to Hagenmaier and Baker [27] with slight modifications [28]. CWN was obtained by inversion phase of the water in oil (W/O) to oil in water (O/W) system in a closed reactor [20]. CWN diameter size obtained was 44.1 ± 7.6 nm, with a narrow polydispersion index (0.28) and zeta potential −43.8 mV, measured by Zetasizer Nano ZS (Malvern Instruments Inc., Westborough, MA, USA) [28]. The incorporation of CEO, as an antimicrobial agent, was done mixing the CEO, 0.75% and 1.50% (*v*/*v*) concentration, and CWN in a high-speed mixer (UltraTurrax T25, IKA Werke GmbH & Co., Staufen, Germany) for 5 min at 5.000 rpm.

CWN and CEO were applied to the fruits randomly divided into four treatments, as follows: Control, CWN (9% of solid phase in suspension), CWN (9%) + CEO (0.75%), CWN (9%) + CEO (1.50%), and non-treated fruits. The coatings were carried out manually by pouring 1 mL of coating solution on latex-gloved hands and manually spread on sanitized papayas. The fruits were stored for 12 days at 16 °C and 70% relative humidity. For replicates, it was 10 fruits for treatment for non-destructive analyses and 10 fruits for destructive, totalizing 80 fruits evaluated.

### 2.3. Non-Destructive Analyses

Papaya weight loss was determined according to the standard method of AOAC [29], by weighing the fruits on day 0 (beginning of the experiment) and on days 3, 6, 9, and 12 of storage. Difference of percentual between the initial and the final weight on each day was used for calculation of weight loss. Skin color was measured with a colorimeter Minolta® CR-400 Chroma Meter (Minolta Camera Co., Osaka, Japan), using the CIELAB system: L* (lightness), a* (green-red), and b* (blue-yellow) values. The hue angle (h°) and chroma (C*) were calculated using Equations (1) and (2), respectively, on the basis of the L*, a*, and b* values (CIELAB color system). In each fruit, three measurements were made in the equatorial region on equidistant sides at the same points throughout the treatment.

$$h^{\circ} = \tan^{-1}\left(\frac{b^*}{a^*}\right) \tag{1}$$

$$C^* = \left((a^*)^2 + (b^*)^2\right)^{1/2} \tag{2}$$

Fruit Severity and Disease Incidence. At the end of the storage period of 12 days at 16 °C, fruits were visually evaluated for incidence and fruit rot severity based on a scale of scores composed of six degrees (0 = absence of symptoms; 1 = 1%–20% affected area; 2 = 21%–40%; 3 = 41%–60%; 4 = 61%–80% and 5 = 81%–100%) [30].

### 2.4. Destructive Analyses

At the end of the storage period of 12 days at 16 °C, the fruit were analyzed. The soluble solids (SS) content was determined using an Atago RX-5000cx digital refractometer (Atago Co., Ltd., Tokyo, Japan), and expressed as %Brix. The pH of the samples was measured using a PHS-3B digital pH meter (3B Scientific., Ltd., Tokyo, Japan) according to the standard method. Titratable acidity was determined using 0.1 M NaOH, with phenolphthalein as the indicator. The results were expressed as g of citric acid per 100 g of fruit. The firmness was evaluated using a digital TA.XTplus Texture Analyser (Stable Micro Systems Ltd., Godalming, UK) with a 6 mm diameter probe, 15 mm s$^{-1}$ velocity, 5 mm penetration distance, and 12 mm$^2$ contact area. The results are expressed in Newtons (N) and the media was calculated based on three penetrations in the distal region of each fruit. All analyses were carried out in triplicate and the data were calculated as means ± standard deviations.

### 2.5. Statistical Analysis

The data are presented as means ± standard deviations. Data were analyzed using analysis of variance (ANOVA) and subsequently by Duncan's multiple amplitude test to assess significant differences between treatments with $p = 0.05$. For severity data, the relative frequency was calculated and the statistical difference, at a 5% significance level ($p < 0.05$), was determined by the Kruskal-Wallis test. Statistical analysis was performed using Sigma Plot 12.0 software.

### 3. Results and Discussion

Weight loss is an important postharvest quality criterion for papaya. Changes to skin permeability and consequently increased water loss are effects of senescence, that can be minimized by coating application [20]. All fruits showed increased weight loss (Figure 1) with increasing storage time, while the lowest weight loss (6.9–7.9%) was obtained for coated papayas. According to Gerreiro et al. [31], edible coatings act as a barrier against moisture loss, leading to a reduction in fruit weight loss. Coatings formulated with CWN and CEO acted as barriers to water diffusion, decreasing fruit transpiration [32]. A similar result was observed for tomatoes [17] and papayas [18] coated with CWN. The coated fruits showed less weight loss compared to the control fruits.

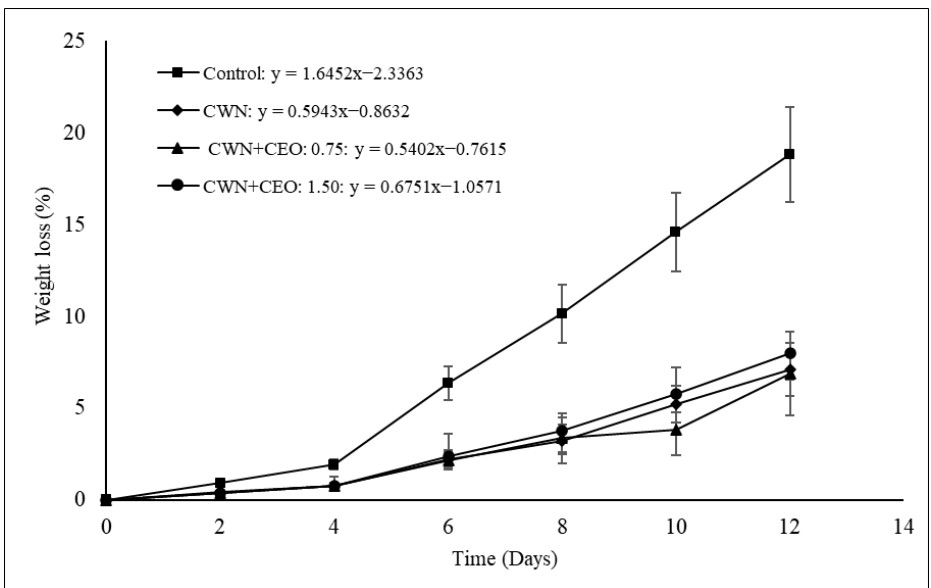

**Figure 1.** Weight loss (%) of control and coated papaya Control: uncoated fruit; CWN: carnauba wax nanoemulsion at 9% of solid phase in suspension; CEO: *C. martinii* essential oil at 0.75% or 1.50%.

Firmness is a determining factor in the postharvest quality of fruits, mainly because it impacts the logistics of the products [32]. Control fruits had the lowest firmness values (Table 1) compared to fruits coated with CWN or CWN + CEO. The smaller loss of firmness observed for coated fruits compared to uncoated may be caused by the reduction of enzymatic activity in the fruit as a consequence of reduced gas exchange and ethylene production by the coating material [33]. The reduction in mass loss may also be related to the decrease in firmness loss in papayas coated with CWN or CWN + OEC during storage. Similar behavior was described by Braga et al. [34] in fruits coated with chitosan and Mentha essential oils.

**Table 1.** Results of physicochemical parameters of papaya uncoated and coated during 12 days of storage (16 ± 1 °C).

| Table | Parameters | | | |
|---|---|---|---|---|
| | Firmness (N) | pH | AT * | SS ** |
| Control | 4.8 ± 0.3 [a] | 5.3 ± 0.2 [a] | 0.11 ± 0.01 [b] | 12.4 ± 0.6 [a] |
| CWN | 5.8 ± 0.7 [b] | 5.6 ± 0.1 [b] | 0.13 ± 0.01 [a] | 11.1 ± 0.6 [b] |
| CWN + CEO 0.75 | 6.3 ± 0.5 [b] | 5.5 ± 0.1 [b] | 0.14 ± 0.01 [a] | 10.9 ± 0.6 [b] |
| CWN + CEO 1.5 | 5.6 ± 0.7 [b] | 5.7 ± 0.2 [b] | 0.14 ± 0.02 [a] | 10.9 ± 0.4 [b] |

Control: uncoated fruit; CWN: carnauba wax nanoemulsion at 9% of solid phase in suspension; CEO: *C. martinii* essential oil at 0.75% or 1.50%. * Titrable acidity (g citric acid equivalent per 100 g of pulp). ** Soluble solids (°Brix). Values in the same column followed by at least one common letter (or not followed by any letters) are not significantly different according to the Duncan's multiple amplitude test ($p < 0.05$).

Fruits coated with CWN or CWN + CEO had higher pH values and lower values for titrable acidity (AT) and soluble solids (SS) values compared to uncoated fruits (Table 1). Higher pH values observed in coated papayas than uncoated may be caused by the decreased use of some organic acids to be converted to sugars in these fruits during storage due to the delay in maturation progress [35]. The results of SS contents also demonstrate that CWN or CWN + OEC based coatings induce a delay in papaya metabolism, reducing the increase in soluble solids [36]. The same behavior has been reported for papaya fruits coated with Whitemouth croaker (*Micropogonias furnieri*) protein isolate and organo-clay nanocomposite [37], chitosan and essential oil of *Ruta graveolens* L. [38], *Aloe vera* [39], and carboxymethylcellulose [40].

One of the most remarkable changes in papaya color during storage is the evolution of the skin color from green to yellow [41]. The results for color analysis of coated and control papayas for 12 days are presented in Table 2. The values of L* for coated fruits were lower when compared to control fruits after the 8th day of storage (Table 2). Chroma values did not generally differ between coated and control fruits during storage. Hue (h°) values decreased in all papayas during storage, being higher for CWN and CWN + CEO when compared to uncoated papaya on days 8 and 12 of storage (Table 2). Similar behavior was described by Braga et al. [34] in fruits coated with chitosan-based edible coatings with Mentha essential oil. The main differences in the C* and h° values of the skin color of uncoated and coated papayas are probably due to the synthesis of carotenoids during ripening [42]. On the other hand, 'Redland' papaya from Florida, coated with CWN at 9%, storage for 10 days at 16 °C followed by 3 days at 22 °C did not show significant differences in Chroma or hue values compared to uncoated fruit; however, higher coating concentration (18%) resulted in color changes delay [18]. The results may be related to variety sensibility. The smaller changes of green to yellow (hue values) in coated fruits compared to uncoated is probably related to the reduction of gas exchange by the coating, which implies a reduction in the enzymatic and chemical reactions involved in the degradation of chlorophyll and/or pigment synthesis [43].

Figure 2A,B shows the results of the effectiveness of the coatings in controlling rots in papayas during storage. Coatings were able to reduce the incidence (Figure 2A) and severity (Figure 2B) of rot in papayas stored at 16 °C for 12 days. The lowest incidence of diseases was observed for fruits coated with CWN + CEO 0.75 and CWN + CEO 1.5, showing a reduction in incidence from 86.70% to 60.00% and 53.33%, respectively. Fruits coated with CWN or CWN+OEC had lower rot severity than control fruits, with 78–89% at 0–2 scores, different from control with 6% at 0–2 scores. Control fruits had the highest severity, 94% of scores 3–5, while coated fruits had only 6–11%.

The addition of the CEO to the CWN coating potentiated the antifungal effect of the coatings (Figure 2B) by reducing the severity of postharvest diseases in papaya fruits. Gonçalves et al. [44] reported that CW inhibited the growth of two phytopathogenic fungi (*Monilinia fructicola* and *Rhizopus tolonifera*) in nectarine and plum. According to Gonçalves et al. [44], the possible antifungal effects of CW may be related to the formation of a stable physical barrier that prevents the passage of the pathogen through the film, along with changes in the internal atmosphere of the fruit and a possible direct antimicrobial action of the wax against pathogens. The antifungal activity of CEO is due to the interactions of its majoritarian compounds geraniol, geranyl acetate, linalool, and caryophyllene [24]. The incorporation of CEO at a concentration of 0.3% in an alginate-based coating was able to improve the microbiological quality of freshly processed melon inoculated with *Salmonella enteritidis* [45]. According to the results, the coating based on CWN and CEO can be considered a promising preservation alternative for papaya, as it inhibited the growth micro-organisms during storage.

**Table 2.** Results of color parameters of papaya uncoated and coated during 12 days of storage (16 ± 1 °C). Results are expressed as average ± standard deviation (n: 10).

| Treatments | Time (Days) | | | | | | | | | | | |
|---|---|---|---|---|---|---|---|---|---|---|---|---|
| | 0 | | | 4 | | | 8 | | | 12 | | |
| | L * | C * | (h°) | L * | C * | (h°) | L * | C * | (h°) | L * | C * | (h°) |
| Control | 59.2 ± 0.0 [a] | 5.1 ± 2.8 [a] | 80.0 ± 1.6 [a] | 67.1 ± 1.5 [a] | 11.2 ± 2.5 [a] | 62.2 ± 2.52 [a] | 62.4 ± 2.2 [b] | 15.0 ± 1.5 [a] | 54.4 ± 3.0 [b] | 49.4 ± 2.1 [b] | 14.6 ± 1.9 [a] | 48.8 ± 2.3 [b] |
| CWN | 61.0 ± 3.6 [a] | 5.7 ± 4.0 [a] | 81.0 ± 2.4 [a] | 66.5 ± 3.8 [a] | 9.0 ± 3.9 [a] | 65.2 ± 3.92 [a] | 68.2 ± 2.1 [a] | 14.1 ± 3.3 [a] | 66.21 ± 4.8 [a] | 56.5 ± 3.8 [a] | 14.6 ± 1.8 [a] | 59.8 ± 2.3 [a] |
| CWN + CEO 0.75 | 60.0 ± 3.8 [a] | 3.8 ± 2.5 [a] | 80.4 ± 2.2 [a] | 67.2 ± 2.9 [a] | 9.4 ± 3.2 [a] | 65.6 ± 3.22 [a] | 68.7 ± 2.2 [a] | 14.0 ± 2.0 [a] | 66.21 ± 3.2 [a] | 58.5 ± 3.0 [a] | 15.0 ± 2.0 [a] | 58.7 ± 2.3 [a] |
| CWN + CEO 1.5 | 55.9 ± 3.9 [a] | 2.8 ± 3.0 [a] | 88.3 ± 2.0 [a] | 67.3 ± 2.4 [a] | 7.8 ± 3.5 [a] | 62.9 ± 3.51 [a] | 66.8 ± 2.0 [a] | 14.4 ± 2.0 [a] | 64.78 ± 3.3 [a] | 55.9 ± 3.6 [a] | 14.5 ± 1.4 [a] | 57.2 ± 3.1 [a] |

* Control: uncoated fruit; CWN: carnauba wax nanoemulsion at 9% of solid phase in suspension; CEO: *C. martinii* essential oil at 0.75% or 1.50%. Values in the same column followed by at least one common letter (or not followed by any letters) are not significantly different according to the Duncan's multiple amplitude test ($p < 0.05$).

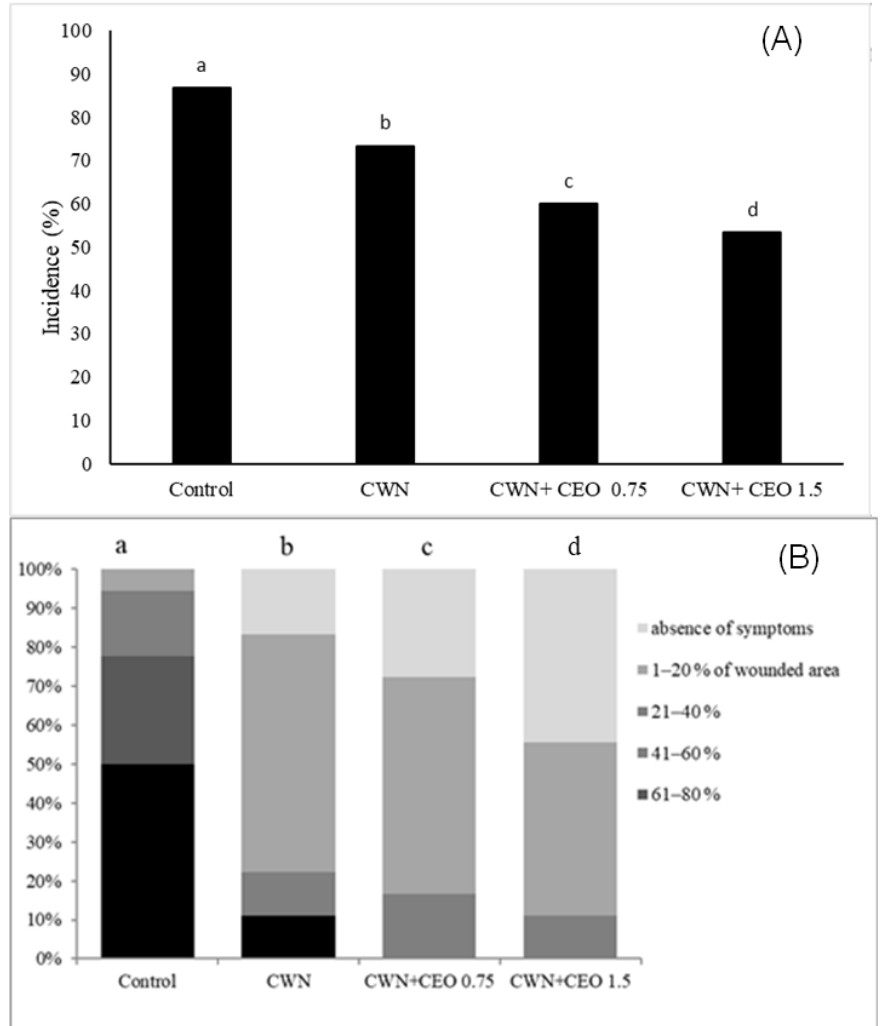

**Figure 2.** Fruit incidence (**A**) and severity (**B**) evaluation at the end of the storage period of 12 days at 16 °C for (a) uncoated, (b) CWN, (c) CWN + CEO 0.75%, and (d) CWN + CEO 1.50%. Control: un-coated fruit; CWN: carnauba wax nanoemulsion at 9% of solid phase in suspension; CEO: *C. martinii* essential oil at 0.75% or 1.50%. Distinct letters represent a significant difference between treatments by the Kruskal–Wallis test ($p < 0.05$).

## 4. Conclusions

Coatings formulated using CWN with or without CEO were applied to papaya fruits and resulted in late ripening during the 12-day storage period, without negatively affecting the overall postharvest quality parameters of the fruit. The coatings reduced weight loss and maintained firmness, in addition to delaying changes in fruit color during storage. Coatings formulated with CWN + CEO, mainly at the highest concentration tested (1.5%), were efficient in reducing the incidence and severity of diseases during fruit storage. The coatings formulated with the incorporation of the CEO presented additional functionalities in the maintenance of the papaya fruit postharvest quality parameters.

**Author Contributions:** Conceptualization, M.D.F.; methodology, J.G.d.O.F., G.d.C.S., F.C.A.O., M.M., C.F., R.M.D.d.O. and M.d.P.G.; investigation, J.G.d.O.F., G.d.C.S., F.C.A.O., M.M., C.F., R.M.D.d.O. and M.d.P.G.; writing—original draft preparation, J.G.d.O.F. and G.d.C.S.; funding acquisition, M.D.F.; supervision, M.D.F.; Writing—review and editing, M.D.F. All authors have read and agreed to the published version of the manuscript.

**Funding:** This research was funded by Embrapa (process 20.19.03.024.00.00), FAPESP (processes 2018/24612-9, 2018/10657-0, 2016/23419-5), CAPES (Grant No. 001), and Brazilian National Council for Scientific and Technological Development-CNPq (process 407956/2016-6), CNPq/MCTI Sisnano (process 442575/2019-0) and M.D.Ferreira CNPq Research Productivity fellowship (process 310728/2019-3).

**Institutional Review Board Statement:** Not applicable.

**Informed Consent Statement:** Not applicable.

**Data Availability Statement:** The authors confirm that the data supporting the findings of this study are available within the article.

**Conflicts of Interest:** The authors declare no conflict of interest.

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
