# Peer review of "Edible Coating Based on Carnauba Wax Nanoemulsion and Cymbopogon martinii Essential Oil on Papaya Postharvest Preservation"

_coatings, doi:10.3390/coatings12111700_

Round 1

Reviewer 1 Report

Coatings-1906744: Edible Coating based on Carnauba Wax Nanoemulsion and Cymbopogon martinii Essential Oil on Papaya Postharvest Preservation

This work is interesting and well conducted, and the manuscript is well written and documented. The experiments are clear, and the results can be applied in food bio-preservation.

However, some questions and suggestions are addressed to the authors :

The effect of essential oil on flavor changes of preserved fruits is not discussed. Please give a comment of this point.

The delay of weight loss observed during the first 4 days of storage (Fig. 1) is not commented. Please could you give an explanation of this phenomenon.

L78 : « previously and published » eliminate “and”.

L153: “the smaller loss of firmness observed for coated fruits…” the loss is “smaller” compared to which reference? Please clarify this point.

L164 (Table 1): the unit of SS (soluble solids) is not given.

L172: “chitosan” instead of “chitosana”

Table 2 : use “point” instead of “comma”.

L176-177: “The values of L* (brightness) and Chroma (color intensity) increased in the on coated and uncoated fruits during storage”. This statement is not totally valid. The results of Table 2 indicate an increase of L values during the first days (4 and/or 8 days) and a decrease in all samples at the last day of storage (day 12). Please verify this point.

L199: change “Figure 2A e B » to “Figure 2A and B”.

Author Response

Revisor #1: This work is interesting and well conducted, and the manuscript is well written and documented. The experiments are clear, and the results can be applied in food bio-preservation.

Response: Thanks for your valuable comments.

The effect of essential oil on flavor changes of preserved fruits is not discussed. Please give a comment of this point.

Response: In this study, unfortunately, the effect of essential oil on papaya flavor during storage was not evaluated. However, some studies have shown that applying coatings functionalized with essential oils has positive effects on flavor chances in papaya fruits. Some of these studies are below:

Miranda, M., Sun, X., Marín, A., Dos Santos, L. C., Plotto, A., Bai, J., ... & Baldwin, E. (2022). Nano-and micro-sized carnauba wax emulsions-based coatings incorporated with ginger essential oil and hydroxypropyl methylcellulose on papaya: Preservation of quality and delay of post-harvest fruit decay. Food chemistry: X, 13, 100249.

de Vasconcellos Santos Batista, D., Reis, R. C., Almeida, J. M., Rezende, B., Bragança, C. A. D., & da Silva, F. (2020). Edible coatings in post-harvest papaya: impact on physical–chemical and sensory characteristics. Journal of food science and technology, 57(1), 274-281.

The delay of weight loss observed during the first 4 days of storage (Fig. 1) is not commented. Please could you give an explanation of this phenomenon.

Response: The delay of weight loss observed during the first 4 days of storage occurred because the fruits were still in the initial stage of ripening. Several studies show that weight loss of papaya increases with the ripening of the fruit, when its metabolism is more active and its rate of respiration is higher.

Miranda, M., Sun, X., Marín, A., Dos Santos, L. C., Plotto, A., Bai, J., ... & Baldwin, E. (2022). Nano-and micro-sized carnauba wax emulsions-based coatings incorporated with ginger essential oil and hydroxypropyl methylcellulose on papaya: Preservation of quality and delay of post-harvest fruit decay. Food chemistry: X, 13, 100249.

de Vasconcellos Santos Batista, D., Reis, R. C., Almeida, J. M., Rezende, B., Bragança, C. A. D., & da Silva, F. (2020). Edible coatings in post-harvest papaya: impact on physical–chemical and sensory characteristics. Journal of food science and technology, 57(1), 274-281.

L78 : « previously and published » eliminate “and”.

Response: It was modified accordingly  (Line 78).

L153: “the smaller loss of firmness observed for coated fruits…” the loss is “smaller” compared to which reference? Please clarify this point.

Response: The comparison was with uncoated fruits. It was modified accordingly (Line 159).

L164 (Table 1): the unit of SS (soluble solids) is not given.

Response: The unit is °Brix.  It was modified accordingly (Line 174).

L172: “chitosan” instead of “chitosana”

Response It was modified accordingly (Line 183).

Table 2 : use “point” instead of “comma”.

Response: It was modified in all table.

L176-177: “The values of L* (brightness) and Chroma (color intensity) increased in the on coated and uncoated fruits during storage”. This statement is not totally valid. The results of Table 2 indicate an increase of L values during the first days (4 and/or 8 days) and a decrease in all samples at the last day of storage (day 12). Please verify this point.

Response: has been modified accordingly (lines 187-189).

L199:  change “Figure 2A e B » to “Figure 2A and B”.

Response: has been modified accordingly (line 209).

Reviewer 2 Report

Author reported edible coating based on carnauba wax nanoemulsion and cymbopogon martinii essential oil on papaya postharvest preservation. The manuscript need to consider the following concerns before it was considered for publication.

1.    The characterization of nanoemulsion should be detailed, for example, the SEM images, the image of particle size, the antibacterial/antioxidant capability.

2.    It was better to show the papaya images during the preservation, which was an intuitive way to observe the color change of papaya.

3.    In the preparation of nanoemulsion, was the weight of the addition of essential  oil based on the  weight of wax? The author should detail the preparing procedure.

4.    It was better to describe the fresh-keeping procedure separately.

Author Response

Reviewer #2: Author reported edible coating based on carnauba wax nanoemulsion and cymbopogon martinii essential oil on papaya postharvest preservation. The manuscript need to consider the following concerns before it was considered for publication.

Response: Thank you for your valuable’s comments.

  1. The characterization of nanoemulsion should be detailed, for example, the SEM images, the image of particle size, the antibacterial/antioxidant capability.

Response: The results of the nanoemulsion characterization were previously published in another article by our research group. Miranda, M., Marilene De Mori, M. R., Spricigo, P. C., Pilon, L., Mitsuyuki, M. C., Correa, D. S., & Ferreira, M. D. (2022). Carnauba wax nanoemulsion applied as an edible coating on fresh tomato for postharvest quality evaluation. Heliyon, 8(7), e09803.

2.It was better to show the papaya images during the preservation, which was an intuitive way to observe the color change of papaya.

Response: The colorimeter is an equipment that provides accurate information about the color of the fruits, allowing the comparison between the results, and it is generally used for postharvest fruit measurements.  More intuitive information could confuse the reader.

  1. In the preparation of nanoemulsion, was the weight of the addition of essential oil based on the weight of wax? The author should detail the preparing procedure.

Response: The concentration of the essential oil was in relation to the final volume of the solution. More details were provided (Line 95).

Reviewer 3 Report

Undoubtedly the topic presented by the authors is very useful. However, their presentation lacks some important points:

·           Fig. 1: Efficiency of the coatings (0.75 < CWN < 1.50) should be discussed. So many decimal figures in lines description is redundant and cannot be substantiated by the experimental measurements, just on the contrary if the experimental errors are considered. Decimal commas should be replaced by decimal points.

·           Table 1: Again more discussion concerning the values of 0.75, CWN and 1.50 should be beneficial. The abbreviations (AT) should be always introduced in the full text.

·           Table 2: The number of the experimental data is not so large. In this respect it seems that the numbers in Table 2 should be rounded to only one decimal figure without any loss of information.

·           Fig. 2A, B: too small letters.

·           Conclusions: No discussion on the differences between 0.75, CWN and 1.50.

Just for improvement:

·           l. 95: Germany

·           l. 96: 5.000?

·           l. 135: test, method

·           l. 175: analysis

·           l. 199: and

·           l. 203: 87 %, 53 %?

·           l. 215: majority?

Author Response

Reviewer #3: Undoubtedly the topic presented by the authors is very useful. However, their presentation lacks some important points:

Response: Thank you for your valuable’s comments.

Fig. 1: Efficiency of the coatings (0.75 < CWN < 1.50) should be discussed. So many decimal figures in lines description is redundant and cannot be substantiated by the experimental measurements, just on the contrary if the experimental errors are considered. Decimal commas should be replaced by decimal points.

Response: The incorporation of C. martinii essential oil as a natural antimicrobial agent did not generally affect the physicochemical properties of the fruits. This behavior was expected since the essential oil was incorporated into the carnauba wax coating as a natural antimicrobial agent. Thus, there was no difference in relation to the physicochemical parameters of the fruits for the two concentrations of essential oil studied. The experimental error has been included in the Figure. Decimal commas were replaced by decimal points.

Table 1: Again more discussion concerning the values of 0.75, CWN and 1.50 should be beneficial. The abbreviations (AT) should be always introduced in the full text.

Response: As previously mentioned, there was no difference in relation to the physicochemical parameters of the fruits for the two concentrations of essential oil studied. In the conclusion, an excerpt was inserted discussing the effect of essential oil concentrations on fruit conservation. The abbreviations (AT) were introduced in the full text. (Line 176).

Table 2: The number of the experimental data is not so large. In this respect it seems that the numbers in Table 2 should be rounded to only one decimal figure without any loss of information.

Response It was modified accordingly.

Fig. 2A, B: too small letters.

Response: It was modified accordingly.

Conclusions: No discussion on the differences between 0.75, CWN and 1.50.

Response: It was modified accordingly (Lines 242-244).

Round 2

Reviewer 2 Report

The manuscript has been revised according to the reviewer's suggestion.

Reviewer 3 Report

The authors reflected majority of my comments.